# Anti-Neuroinflammatory Effects of a Macrocyclic Peptide-Peptoid Hybrid in Lipopolysaccharide-Stimulated BV2 Microglial Cells

**DOI:** 10.3390/ijms25084462

**Published:** 2024-04-18

**Authors:** Lu Sun, Soraya Wilke Saliba, Matthias Apweiler, Kamil Akmermer, Claudine Herlan, Christoph Grathwol, Antônio Carlos Pinheiro de Oliveira, Claus Normann, Nicole Jung, Stefan Bräse, Bernd L. Fiebich

**Affiliations:** 1Neuroimmunology and Neurochemistry Research Group, Department of Psychiatry and Psychotherapy, Medical Center-University of Freiburg, Faculty of Medicine, University of Freiburg, D-79104 Freiburg, Germany; 2Institute of Organic Chemistry, Karlsruhe Institute of Technology (KIT), D-76131 Karlsruhe, Germany; kamil.akmermer@kit.edu (K.A.); claudine-herlan@web.de (C.H.); braese@kit.edu (S.B.); 3Institute of Biological and Chemical Systems-Functional Molecular Systems (IBCS-FMS), Karlsruhe Institute of Technology (KIT), Kaiserstrasse 12, D-76131 Karlsruhe, Germany; 4Neuropharmacology Laboratory, Department of Pharmacology, Universidade Federal de Minas Gerais, Belo Horizonte 31270-901, Brazil; 5Mechanisms of Depression Research Group, Department of Psychiatry and Psychotherapy, Medical Center-University of Freiburg, Faculty of Medicine, University of Freiburg, D-79104 Freiburg, Germany

**Keywords:** neuroinflammation, microglia, depression, peptoid, cytokines, chemokines

## Abstract

Inflammation processes of the central nervous system (CNS) play a vital role in the pathogenesis of several neurological and psychiatric disorders like depression. These processes are characterized by the activation of glia cells, such as microglia. Clinical studies showed a decrease in symptoms associated with the mentioned diseases after the treatment with anti-inflammatory drugs. Therefore, the investigation of novel anti-inflammatory drugs could hold substantial potential in the treatment of disorders with a neuroinflammatory background. In this in vitro study, we report the anti-inflammatory effects of a novel hexacyclic peptide-peptoid hybrid in lipopolysaccharide (LPS)-stimulated BV2 microglial cells. The macrocyclic compound X15856 significantly suppressed Interleukin 6 (IL-6), tumor necrosis factor-α (TNF-α), c-c motif chemokine ligand 2 (CCL2), CCL3, C-X-C motif chemokine ligand 2 (CXCL2), and CXCL10 expression and release in LPS-treated BV2 microglial cells. The anti-inflammatory effects of the compound are partially explained by the modulation of the phosphorylation of p38 mitogen-activated protein kinases (MAPK), p42/44 MAPK (ERK 1/2), protein kinase C (PKC), and the nuclear factor (NF)-κB, respectively. Due to its remarkable anti-inflammatory properties, this compound emerges as an encouraging option for additional research and potential utilization in disorders influenced by inflammation, such as depression.

## 1. Introduction

Depression is a complex and debilitating mental disorder, with a high prevalence among psychiatric disorders, affecting all genders and ages worldwide [1]. Current pharmacological therapies for depression exert many side effects [2,3] and around 30% of the patients have only partial or no improvement in the symptoms using different available antidepressants [4,5]. This may reflect the fact that the pathogenesis of depression is still not completely understood. Beyond the imbalance of neurotransmitters in the central nervous system (CNS), studies suggest a relevant role of neuroinflammation in the pathophysiology of depression [1,6,7].

Chronic neuroinflammation is characterized by the activation of the immune cells of the CNS, mainly glial cells, and the release of proinflammatory cytokines and chemokines that may contribute to a range of symptoms, including a depressed mood, a lack of motivation, and altered cognition [8]. Microglial cells are the resident macrophages of the CNS and are distributed in several brain areas, including the cortex and hippocampus, which are affected by depression. These cells represent approximately 10% of the CNS cells in a healthy brain and are characterized by a ramified morphology and small cellular bodies. In response to an injury or any disturbance in the brain environment, they modify their morphology and the cells exhibit rapid and profound changes in the cell shape, gene expression, and functional behavior, characterized by the reduction in cellular processes and the development towards an amoeboid appearance [9,10]. Microglial cells are crucial for homeostatic state maintenance, being involved in starting, amplifying, balancing, and solving the inflammatory response [11], and are involved in other processes like monitoring synaptic function, phagocytosis, and regulating synaptic plasticity [12,13]. Studies have shown a correlation between microglial impairment in either function or structure and depression. Therefore, some authors have interpreted selected cases of depressive disorder as a microglial disease or microgliapathy [14,15,16].

We showed different approaches for the modulation of neuroinflammation using different compounds in LPS-stimulated microglial cells [17,18]. These compounds exert anti-inflammatory effects by decreasing the release or expression of prostaglandins, cytokines such as interleukin-6 (IL-6), or tumor necrosis factor (TNF)-α. The anti-inflammatory effects are partially mediated by the prevention of the activation of mitogen-activated kinases (MAPK) signaling, such as c-Jun N-terminal Kinase (JNK), Extracellular Signal-Regulated Kinase (ERK), and p38 MAPK [19].

In this study, we investigated the effects of a cyclic peptide-peptoid hybrid in LPS-stimulated BV2 microglial cells [20]. Peptoids are a class of molecules that mimic the biological interactions of peptides (peptidomimetics) but have high stability against proteases, high cellular permeability, and reduced immunogenicity as compared to the corresponding peptide [21]. Furthermore, peptoid units provide a significantly higher degree of side-chain variability compared to their natural amino acid counterparts. However, this modification is also accompanied by the loss of stereo information at the α-carbon atom of the amino acid motif, which is typically crucial for specific and profound ligand-receptor interactions [22]. Peptide-peptoid hybrids are therefore designed to harness the positive properties of both substance classes, aiming to achieve both high affinity and selectivity towards the biological target, as well as an improved metabolic and immunogenic profile of the drug [20].

Studies have shown novel therapeutic opportunities of peptidomimetics modulating cell signaling pathways [23,24], neuroinflammation [25,26], oxidative stress [27], apoptosis [28,29], and depression [21,30,31]. For example, peptoids such as RAGEs (receptor for advanced glycation end-products) antagonists can modulate the RAGE-induced inflammatory response [26]. A structure-based development of new peptidomimetic cyclic compounds was conducted for the PSD-95 PDZ3 domain, which is a promising therapeutical target for depression [32]. These studies demonstrate the involvement of distinct peptidomimetics in different processes, suggesting peptide-peptoid hybrids as an interesting approach to modulate neuroinflammation in different contexts, including depression.

For the current study, the hexacyclic peptide-peptoid hybrid X15856 was synthesized for the first time, and its effects on gene expression and the protein release of chemokines and cytokines induced by LPS in BV2 microglial cells were investigated. Furthermore, the effects on different cell signaling pathways were studied.

## 2. Results

### 2.1. Effects of X15856 on Cell Viability

The cytotoxic effects of X15856 (1-25 μM) in BV2 microglial cells were ruled out using an MTT assay. As shown in Figure 1, X15856 did not exert cytotoxic effects in LPS-stimulated BV2 microglial cells. The solvent of the compound, DMSO 0.1%, did not affect cell viability either, whereas 20% of Ethanol, as a positive control, significantly induced cell death. Since X15856 did not affect cell viability, it was used for further experiments in concentrations up to 25 μM.

### 2.2. Effects of X15856 on Gene Expression and Protein Synthesis of TNF-α

Since TNF-α is a vital proinflammatory cytokine in the CNS [33] and different studies have demonstrated its role in depression [34,35,36]; we measured the effects of X15856 on gene expression and the protein synthesis of TNF-α in LPS-stimulated BV2 microglial cells. Briefly, BV2 microglial cells were pre-treated with X15856 (1 μM to 25 μM) for 30 min, followed by the stimulation of gene expression and protein synthesis via 4 or 24 h treatment with LPS (10 ng/mL). The expression and synthesis of TNF-α is reliably induced by LPS. X15856 showed a significant and concentration-dependent decrease of LPS-induced TNF-α expression and synthesis, starting at concentrations of 10 μM in both gene expression (Figure 2A) and protein synthesis (Figure 2B). X15856 reduced TNF-α release to baseline levels in the highest concentration of 25 μM and demonstrated a significant decrease starting at 10 μM.

### 2.3. Effects of X15856 on Gene Expression and Protein Synthesis of IL-6

As numerous research studies outlined the associations between depression and IL-6 [37,38], we measured the expression and release of IL-6 in LPS-treated BV2 microglial cells. Microglia was pre-incubated with X15856 (1 μM to 25 μM) for 30 min. Afterwards, LPS was added for 4 or 24 h. The transcriptional regulation of IL-6 with X15856 in BV2 microglial cells was measured with qPCR and the release of IL-6 was determined in the cell supernatants which were collected after centrifugation. The gene expression and protein synthesis of IL-6 were both strongly induced by LPS stimulation (Figure 3). Pre-treatment with X15856 showed a comparable significant and concentration-dependent reduction to baseline levels of untreated cells for both gene expression (Figure 3A) and protein synthesis (Figure 3B).

### 2.4. Effects of X15856 on Gene Expression and Protein Synthesis of CCL2

Chemokines, such as the CCL2 (small chemokine from the CC-chemokine family), play a vital role in neuropsychiatric disorders, such as depression [36,39]. The stimulation of microglial cells with LPS (10 ng/mL) for 4 or 24 h caused a significant increase in the gene expression and protein synthesis of CCL2, respectively, as shown in Figure 4. Pre-treatment with X15856 led to the strong inhibition of gene expression starting at concentrations of 1 μM. However, baseline gene expression was not reached at the highest concentration of 25 μM (Figure 4A). Furthermore, X15856 inhibited CCL2 release significantly and concentration-dependently (Figure 4B). X15856 did not reach baseline CCL2 release in its highest concentration of 25 μM but showed a reduction to 50% of the LPS positive control.

### 2.5. Effects of X15856 on Gene Expression and Protein Synthesis of CCL3

CCL3 is a small chemokine belonging to the CC-chemokine family and is one of the most upregulated chemokines during depression [39,40]. LPS stimulation reliably increased CCL3 gene expression and release in BV2 microglial cells (Figure 5). X15856 exerted a concentration-dependent inhibition of LPS-induced CCL3 gene expression and release. CCL3 gene expression was strongly reduced to almost baseline expression in the highest concentration of 25 μM (Figure 5A). The compound furthermore suppressed CCL3 release starting at concentrations of 1 μM (Figure 5B) beyond basal levels at 25 μM.

### 2.6. Effects of X15856 on Gene Expression and Protein Synthesis of CXCL2

CXCL2, also recognized as macrophage inflammatory protein-2 alpha (MIP-2 α), growth-regulated protein beta (GRO-β), and growth oncogene-2 (Gro-2), is a small chemokine expressed in various cells, such as microglia, astrocytes, and neurons and plays an important role in inflammation [41] and depression [42]. LPS stimulation reliably increased the CXCL2 expression and release in BV2 microglial cells (Figure 6). X15856 exerted a significant and concentration-dependent inhibition of LPS-induced CXCL2 gene expression and protein synthesis in BV2 microglial cells. X15856 showed a concentration-dependent reduction in CXCL2 gene expression starting at concentrations of 1 μM (Figure 6A) and a significant inhibition of CXCL2 release starting at concentrations of 10 μM, not reaching basal CXCL2 levels at 25 μM (Figure 6B).

### 2.7. Effects of X15856 on Gene Expression and Protein Synthesis of CXCL10

CXCL10, also called interferon-inducible protein (IP-10), promotes chemotaxis and modulates the inflammatory responses of cells [43] contributing to depression [44,45,46]. The LPS stimulation of BV2 microglial cells significantly upregulated the expression and release of CXCL10. X15856 showed a concentration-dependent inhibition of LPS-stimulated CXCL10 gene expression and protein synthesis. A significant inhibition of CXCL10 gene expression started at concentrations of 10 μM (Figure 7A) and a significant inhibition of CXCL10 release from a concentration of 5 μM (Figure 7B).

### 2.8. Comparison of X15856 to Dexamethasone and Hydrocortisone

We further compared the results obtained with X15856 with the effects of the well-known anti-inflammatory drugs dexamethasone (Appendix A) and hydrocortisone (Appendix A) on the many parameters in LPS-stimulated BV2 microglial cells. As expected, both reference drugs potently inhibited LPS-induced cytokine and chemokine release in the concentrations of 0.1, 1, and 10 μM.

The synthesis of CXCL2 was more potently inhibited by X15856 when compared to dexamethasone or hydrocortisone; comparable effects were achieved with CCL3.

### 2.9. Effects of X15856 on Phosphorylation of PKCβ, p38 MAPK, ERK ½, and NF-κB

The MAPK signaling pathway, which includes p38 MAPK and ERK, regulates various cellular activities. Accumulating evidence indicates the important role of MAPK signaling in brain inflammation and degeneration [47,48,49]. In this study, we investigated the phosphorylation of ERK 1/2 and p38 MAPK in LPS-stimulated BV2 microglial cells.

As shown in Figure 8, LPS stimulation induced the phosphorylation of ERK 1/2 and p38 MAPK. Pre-treatment with X15856 for 30 min inhibited the LPS-induced phosphorylation of ERK1/2 and p38 MAPK. The compound showed a concentration-dependent inhibition pattern and achieved a significant decrease in ERK 1/2 phosphorylation at the highest concentration (25 μM) (Figure 8A). Furthermore, X15856 demonstrated a significant decrease in p38 MAPK phosphorylation at concentrations of 10 and 25 μM (Figure 8B).

PKC is a family of phospholipid-dependent serine/threonine kinase and is classed into three subfamilies based on their structural and activation characteristics. The antibody used in this study is designed to detect phosphorylation at the Ser660 site of PKC βII. Consequently, it also detects PKC α, βI, βII, δ, ε, η, and θ isoforms when phosphorylated at the same site. As shown in Figure 9A, LPS stimulation induced the phosphorylation of PKC (pan) (βII Ser660) and X15856 significantly reduced PKC (pan) (βII Ser660) phosphorylation at concentrations of 10 and 25 μM.

NF-κB is a family of dimeric transcription factors central to coordinating inflammatory responses. Hence, we studied the effects of X15856 on the phosphorylation of NF-κB. NF-κB phosphorylation was strongly induced by LPS and X15856 showed a concentration-dependent inhibition of NF-κB phosphorylation reaching significance at the highest concentration of 25 μM (Figure 9B).

## 3. Discussion

In this study, we further investigated the effects of the novel synthesized peptoid X15856 on cytokines and chemokines. The novel compound showed significant suppression of the gene expression and protein synthesis of the cytokines IL-6 and TNF-α, as well as the chemokines CCL2, CCL3, CXCL2, and CXCL10. The results suggest a dose-response effect of X15856 on both protein and RNA levels. Furthermore, we demonstrated that the anti-inflammatory effects might be partially dependent on the PKC/MAPK signaling pathway and NF-κB signaling. Overall, X15856 exerts promising anti-inflammatory effects in BV2 microglial cells. Specifically, the synthesis of CXCL2 was more potently inhibited by X15856 when compared to dexamethasone or hydrocortisone; comparable effects were obtained on LPS-induced CCL3 synthesis.

A wealth of knowledge indicates the pivotal role of microglia in neuroinflammatory processes. Microglia are CNS-resident myeloid cells that originate from common erythron-myeloid progenitors (EMPs), which are important inflammatory reaction cells involved in the pathogenesis of neuropsychiatric disorders, such as Alzheimer’s disease [50], Parkinson’s disease [51], and depression [15]. There is a growing interest in developing drugs and therapies that specifically target microglia to control and regulate neuroinflammatory processes [52,53]. Furthermore, proinflammatory cytokines play an important role in the mediation of depressive disorders [54]. According to a recent meta-analysis of cytokines in major depression, the proinflammatory cytokines IL-6 and TNF-α were significantly increased in depressed subjects when compared to healthy controls [55]. It has been reported that more than half of MDD (Major Depression Disorder) patients have associated comorbidities that are often associated with microglial inflammation [56]. Previous studies have reported that the experimental activation of microglia exacerbated the development of depressive-like behaviors in animal models [57] and antidepressant drugs were shown to alleviate the activation of microglia [58,59]. The glucagon-like peptide-1 (GLP-1) receptor agonists and dipeptidyl peptidase-IV (DPP-IV) inhibitors showed anti-inflammatory properties in humans and exerted a reduction in the depressive symptoms of newly diagnosed type 2 diabetic patients with major depression, suggesting that a reduction in inflammation might provide antidepressant effects [60]. The peptoid-related approach for developing novel diagnostic and therapeutic strategies in neuropsychiatric disorders is gaining interest in research. The peptoid PD2 showed a high accuracy in identifying de novo Parkinson’s disease (PD), which may be useful for the early-stage identification of PD [61]. Furthermore, compared to peptide-based drugs, cyclic peptoids have favorable advantages, such as increased stability, enhanced binding affinity, improved bioavailability, and reduced conformational flexibility [62,63]. However, there is a lack of research focusing on peptoid-related drugs in depression. Due to their advantages compared to peptides, the development of novel peptoid-based drugs might offer new therapeutic options and strategies.

Peptoids are peptidomimetics and sequence-specific *N*-substituted glycine oligomers that evade proteolytic degradation via the repositioning of the side chain from the α-carbon to the amide nitrogen. Therefore, peptoids are isomerically related to peptides but their side chains are attached to the backbone amide nitrogens instead of the backbone α-carbon [21]. This shift has a major impact on the conformational flexibility of the spatial structure of peptoids. The synthesis of peptoids is possible at a comparatively low cost, with improved biostability and bioavailability, and reduced immunogenicity compared to similar peptides [64,65]. Like other unnatural peptidomimetics, peptoids are proteolytically stable and are known to have better membrane permeability than native peptides [22]. As research has demonstrated, cyclic peptoids exhibit relatively rigid and preorganized structures compared to linear peptoids. This characteristic leads to a higher affinity to target proteins and reduces the conformational flexibility of cyclic peptoids [66]. A reduced target binding affinity of flexible and disorganized molecular structures due to the loss of conformational entropy and loss of motional freedom to fit the target binding pocket is known as entropic penalty [67]. Therefore, rigid and highly preorganized binding sites of molecules/ligands, such as cyclic peptoids, are associated with mitigated entropic penalty explaining the enhanced specificity and higher target binding affinities [66].

It was also reported that cyclic peptoids are far more cell-permeable than their linear counterparts irrespective of their size and side chains [62]. Furthermore, the incorporation of a peptoid unit into a series of peptidic macrocycles led to high-affinity CXC chemokine receptor 7 (CXCR7) ligands with increased cell permeability [68]. CXCR7 is a G-protein-coupled receptor and can interact with either CXCL11 or CXCL12 that has been implicated in various aspects of inflammation and psychiatric disorders [69]. For example, chronic fluoxetine treatment showed a normalization of the central nervous CXCL12-CXCR4-CXCR7 axis in a prenatal stress-induced depression animal model [70]. Furthermore, CXCR7 expression is increased in the airway epithelium and involved in the regulation of allergic airway inflammation [71]. In the current study, we investigated the anti-inflammatory effects of one novel peptoid, X15856, which is a cyclic peptoid, in BV2 microglial cells. X15856 demonstrated significant inhibition of LPS-induced cytokine (IL-6 and TNF-α) and chemokine (CCL2, CCL3, CXCL2, and CXCL10) release.

Therefore, we further investigated pathways possibly involved in cytokine and chemokine release to understand the molecular mechanisms responsible for the anti-inflammatory effects of X15856. It has been shown that PKC/MAPK and NF-κB signaling is involved in inflammatory processes. These pathways play crucial roles in the regulation of immune responses, cell proliferation, and the production of inflammatory mediators [72,73,74]. Therefore, the inhibition of the phosphorylation of those signal pathway members might be responsible for reduced cytokine and chemokine expression and release. In line with the previous study of our group, LPS induced the phosphorylation of MAPK, PKC (pan) (βII Ser660), and NF-κB in LPS-stimulated BV2 cells [75]. PKC, as a phospholipid-dependent serine/threonine kinase, appears to be involved in the signal transduction response to diverse biological phenomena and diseases, such as depression [76]. In the presence of Ca^2+^ influx, PKC mediates the activation of the IκB kinase (IKK) complex and NF-κB pathway [77]. PKC-regulating peptides or peptidomimetics stimulate or inhibit the resulting signaling pathways in a highly specific manner, which is necessary to generate specific therapeutic agents [78]. There are 11 different PKC isozymes responsible for distinct cell responses [79]; however, this study only evaluated the effects on the PKC βII Ser660 phosphorylation site due to the antibody used. Peptides and peptoids might be designed to selectively interact with specific PKC isoforms and this selectivity might allow for the fine-tuning of PKC-mediated signaling in a cellular context. Another peptoid ligand of the fibroblast growth factor receptors (FGFRs) activated FGFR signaling pathways, leading to an increase in ERK phosphorylation in different cell lines [80]. Generally, peptoids might possibly be used as alternatives to peptides in various cellular processes and could be potentially applied in drug development.

Depression stands out as one of the most common severe psychiatric illnesses. Inflammatory processes have been suggested as part of the pathophysiology of depression, and the suppression of neuroinflammation has emerged as a promising focus for therapeutic advancements. Growing evidence indicates that peptoids are advantageous in neurotherapeutic applications because of their anti-inflammatory effects. Peptoids have been suggested as a means of modulating antigen-specific immune responses by acting as “antigen surrogates”, which enhance a weak immune response or attenuate a harmful immune response through competitive interactions [81]. It is reported that peptoid mimics of catalase and superoxide dismutase were used to suppress oxidative stress, which is an integral component of inflammation [82]. In order to modulate the activation of immune cells, lipidated peptide-peptoid hybrids have been identified as antagonists acting on the formyl peptide receptor (FPR) 2, which is expressed by human neutrophils. After binding FRP2, these antagonists are able to attenuate the FRP2-specific activation of neutrophils [83]. In another study about lipidated peptide/β-peptoid hybrids, proteolytically stable host defense peptides (HDP) mimetics displaying anti-inflammatory activity and formyl peptide receptor 2 antagonism in human and mouse immune cells in vitro showed attenuated phorbol 12-myristate 13-acetate (PMA)-induced ear edema and reduced the local production of the proinflammatory chemokines MCP-1 and CXCL-1 and the cytokine IL-6 [84]. Furthermore, some novel peptoids exhibited axonal and myelin neuroprotection in brain inflammation. BN201, screened from combinatorial libraries, displays an array of neuroprotective effects in neurons and myelin-forming cells and is currently being developed as a neuroprotective therapy for MS [85]. It was also reported that peptoids can be specifically designed to mimic the hydrophobic core of Aβ in AD and incorporate a neutral, positive, or negative spacer between aromatic side chains, which can abate Aβ aggregation and reduce the aggregate-induced NF-κB activation in neuronal cells [86]. Therefore, peptoids can be designed and synthesized to mimic the structure and function of natural peptides, and the anti-inflammatory characteristic and possibly milder side effects might be an alternative to the already known anti-inflammatory drugs. However, the pharmacological use of peptoids is not yet established. Our results show promising anti-inflammatory effects of the tested peptoid X15856, but further experiments are necessary to investigate further pathways and mechanisms involved in its effects. Additionally, in vivo experiments are necessary to confirm the effects in a complex organism and to reveal possible side effects.

In summary, we provide evidence that X15856, as a novel cyclic peptoid, has significant anti-inflammatory effects partially dependent on the MAPK signaling pathway, PKC signaling pathway, and NF-κB signaling pathway. Its anti-inflammatory properties might be beneficial in inflammation-associated diseases.

## 4. Materials and Methods

### 4.1. Chemistry

#### 4.1.1. General Aspects

Reagents (ABCR GmbH, Karlsruhe, Germany; Thermo Fisher Scientific, Waltham, MA, USA; Carbolution Chemicals GmbH, St. Ingbert, Germany; Chempur Feinchemikalien und Forschungsbedarf GmbH, Karlsruhe, Germany; Merck Millipore, Merck Novabiochem and Sigma-Aldrich, Merck KGaA, Darmstadt, Germany; Carl Roth GmbH & Co. KG, Karlsruhe, Germany; TCI Europe N.V., Zwijndrecht, Belgium; VWR International GmbH, Darmstadt, Germany) were commercially purchased and used without further purification. Absolute or HPLC-Gradient-Grade solvents were commercially acquired (ABCR GmbH, Karlsruhe, Germany, Thermo Fisher Scientific, Waltham, MA, USA and VWR International GmbH, Darmstadt, Germany) and employed without further purification. Solvent mixtures were measured volumetrically. Water was deionized with an Elix^®^ Essential 15 water purification system (Merck Millipore, Merck KGaA, Darmstadt, Germany) and additionally treated for chromatographic applications using a Milli-Q Biocel system (Q-Gard^®^ 1, Quantum^®^ X, Merk Millipore, Merck KGaA, Darmstadt, Germany). Solutions of inorganic salts, unless otherwise specified, were used as saturated aqueous solutions.

Solvents, unless otherwise indicated, were removed at a 40 °C bath temperature using a rotary evaporator under reduced pressure. Aqueous solutions containing the product were frozen with liquid nitrogen at −196 °C and freeze-dried under reduced pressure using a Lyophilizer (LDC-1, Alpha 2-4, Martin Christ Gefriertrocknungsanlagen GmbH, Osterode am Harz, Germany).

The weighing of different substances was performed on the analytical balance of type CP224S (d = 0.1 m, Sartorius AG, Göttingen, Germany) or the model AS 220.X2 (d = 0.1 mg, Radwag UK Ltd., Macclesfield, United Kingdom).

During solid-phase synthesis, an orbital shaker type KS501 Digital (IKA-Labortechnik, IKA-Werke GmbH & CO. KG, Staufen, Germany) was used for mixing.

For the purification, the Puriflash™ 4125 preparative High-Performance Liquid Chromatography (HPLC) system (Advion Interchim Scientific, Montluçon, France) was employed, along with InterSoft^®^ V5.1.08 software. Chromatographic separation was conducted using the VDSpher^®^ C18-M-SE guard column (10 μm, 40 × 16 mm) and the VDSpher^®^ C18-M-SE separation column (10 μm, 250 × 20 mm, VDS optilab Chromatographietechnik GmbH, Berlin, Germany). For the analysis, the Ultimate™ 3000 Liquid Chromatography-Mass Spectrometry system (Thermo Fisher Scientific, Waltham, MA, USA) was employed, utilizing Chromeleon 7.2.10 ES software. The chromatographic separation was carried out using the VDSpher^®^ CSM-C-18-M-SE separation column (2.5 μm, 100 × 2 mm, VDS optilab Chromatographietechnik GmbH, Berlin, Germany). The system components included the UltiMate™ 3000 Series SD, BM, and RS Pump Series for pumping, the UltiMate™ 3000 Series ACC-3000 Autosampler Column Compartment for sample injection, and the UltiMate™ 3000 Series DAD-3000(RS) and MWD-3000(RS) Diode Array Detectors for UV detection (Thermo Fisher Scientific, Waltham, MA, USA). The mass spectrometric analysis was performed using the ISQ™ EM Single Quadrupole Mass Spectrometer (Thermo Fisher Scientific, Waltham, MA, USA).

#### 4.1.2. Synthesis of cyclo-(L-Phe-N1phpCl-N1ph-N1phpCl-N4am-L-Phe); X15856

In a polypropylene syringe with a polyethylene frit and cannula (Item number: V050PE063; MULTISYNTECH GmbH, Witten, GERMANY), 2-chlorotritylchloride polystyrene resin (125 mg, 200 μmol, 1.60 mmol/g loading density, 100–200 mesh, 1.00 equiv.) was swollen for at least 30 min at 21 °C in dichloromethane (DCM) using an orbital shaker. The solvent was removed, and for the coupling of the first amino acid Fmoc-L-phenylalanine (310 mg, 800 μmol, 4.00 equiv.), dissolved in 2.00 mL of *N*-methylpyrrolidone (NMP) and *N*,*N*-diisopropylethylamine (DIPEA, 139 μL, 103 mg, 800 μmol, 4.00 equiv.) was added to the resin. The reaction mixture was shaken overnight at 21 °C. After removing the solvent, it was washed with 2.00 mL each of *N*,*N*-dimethylformamide (DMF), methanol, DCM, and DMF once more.

To cleave the Fmoc-protecting group, the loaded resin was treated with 2.50 mL of a 20% piperidine solution in DMF and shaken for 5 min at 21 °C. The solution was removed, and the process was repeated twice. Subsequently, the resin was washed with 2.00 mL each of DMF, methanol, DCM, and DMF once more.

For the acetylation, the loaded resin was treated with 1.60 mL of a 1.00 M solution of bromoacetic acid (222 mg, 1.60 mmol, 8.00 equiv.) in DMF, with diisopropylcarbodiimide (DIC, 250 μL, 202 mg, 1.60 mmol, 8.00 equiv.) and incubated for 30 min at 21 °C. The reaction mixture was removed, and the resin was washed with 2.00 mL each of DMF, methanol, DCM, and DMF once more.

For the substitution, *p*-chlorobenzylamine (195 μL, 1.60 mmol, 8.00 equiv.), dissolved in 2.00 mL of DMF, was added to the resin, and incubated for 1 h at 21 °C. The solution was removed, and the resin was washed with 2.00 mL each of DMF, methanol, DCM, and DMF once more.

Acetylation and substitution with benzylamine (174 μL, 171 mg, 1.60 mmol, 8.00 equiv.), *p*-chlorobenzylamine (195 μL, 227 mg, 1.60 mmol, 8.00 equiv.), and *N*-Boc-diaminobutane (310 mg, 1.60 mmol, 8.00 equiv.) were repeated.

Subsequently, Fmoc-L-phenylalanine (310 mg, 800 μmol, 4.00 equiv.) was coupled by dissolving it in 2.00 mL of NMP along with 1-hydroxybenzotriazole (HOBt, 123 mg, 800 μmol, 4.00 equiv.) and treating the solution with DIC (125 μL, 101 mg, 800 μmol, 4.00 equiv.). The reaction mixture was then added to the resin and agitated for 4 h at 21 °C. After the coupling solution was removed, the resin was washed with 2.00 mL of DMF, methanol, DCM, and DMF once more.

For the cleavage of the resin, the resin was washed three times with 2.00 mL of DCM and then treated with 2.50 mL of a 33% solution of hexafluoroisopropanol (HFIP) in DCM for two rounds of 1 h at 21 °C. The cleavage solution was collected, and the resin was washed five times with DCM. The solvent was removed by a counter-flow of air, and the residue was dissolved in acetonitrile and water and subsequently lyophilized.

The obtained crude linear precursor was used in the final cyclization step without further purification:

*N*,*N*,*N*′,*N*′-tetramethyl-O-(7-azabenzotriazol-1-yl)uroniumhexafluorophosphate (HATU, 38.78 mg, 102 μmol, 0.750 equiv.) was dissolved in 30.0 mL of DMF. The linear precursor (147.8 mg, 136 μmol, 1.00 equiv.) was dissolved in 20.0 mL of DMF and mixed with DIPEA (189 μL, 8.00 equiv.). In total, 10 mL of this solution was added slowly (1.60 mL/h) to the HATU solution in DMF at 21 °C. Subsequently, another portion of HATU (0.750 equiv.) was added in one portion to the reaction mixture. The remaining 10 mL of the linear precursor solution was added dropwise (1.60 mL/h), and the mixture was stirred at 21 °C for 12 h. The solvent was removed under reduced pressure.

To remove the Boc-protecting group, the obtained residue was treated with 5.00 mL of a 95% solution of trifluoroacetic acid (TFA) in DCM and stirred for 1 h at 21 °C. The solvent was removed by a counter-flow of air.

The purification of the residue by preparative HPLC (5-95% acetonitrile in water, with 0.1% TFA over 45 min) and successive lyophilization yielded the product as a colorless powder (23.3 mg, 25.0 μmol, 18% yield over 15 reaction steps).

LC-MS (5-95% acetonitrile + 0.1% HCOOH in 10 min, detection at 214 nm): t_Ret_ = 4.86 min (91%); HRMS (ESI, C_51_H_55_Cl_2_N_7_O_6_): calc.: 932.3661 [M+H]^+^; found: 932.3669.

#### 4.1.3. Usage of X15856 and Other Chemicals

X15856 (Figure 10) was synthesized by the Karlsruher Institute for Technology (KIT), Institute of Organic Chemistry, dissolved as 10 mM stocks, dissolved in DMSO (Merck KGaA, Darmstadt, Germany), and used in final concentrations of 1, 5, 10, and 25 μM. A total of 5 mg/mL lipopolysaccharide (LPS) from Salmonella typhimurium (Sigma Aldrich, Deissenhofen, Germany) was dissolved in PBS as stock solution and diluted with distilled water for a final concentration of 10 ng/mL in BV2 microglial cultures [87]. The anti-inflammatory drugs dexamethasone (Cayman Chemical distributed by Biomol, Hamburg, Germany) and hydrocortisone (Cayman Chemical distributed by Biomol, Hamburg, Germany) were dissolved in DMSO and used in final concentrations of 0.1, 1, and 10 μM.

### 4.2. BV2 Microglial Cell Culture

The BV2 microglial cells were kindly provided by Prof. Langmann (Department of Ophthalmology, University of Cologne, Cologne, Germany) and cultured in 1× RPMI 1640 medium containing 10% fetal calf serum (FCS, Bio and SELL GmbH, Feucht/Nürnberg, Germany), 2 mM of L-glutamine, and 1% penicillin/streptomycin (all cell culture solutions obtained by Gibco, Thermo Fisher Scientific, Bonn, Germany) under a 5% CO_2_, 37 °C, and humidified culture atmosphere. When approx. 90% confluency was reached, cells were passed by trypsin and reseeded to 6-, 12-, 24-, or 96-well plates or new cell culture flasks, respectively. On the next day, the medium was changed, and after 1 h, the cells were stimulated for respective experiments.

### 4.3. Cell Viability Assay

An MTT assay (Sigma-Aldrich GmbH, Taufkirchen, Germany) was performed to identify the possible cytotoxic effects of X15856 or the vehicle. This assay measures the number of metabolically active cells and allows conclusions about viable cells in the culture, determined by the reduction of a yellow tetrazolium salt (3-(4,5-dimethylthiazol-2-yl)-2,5-diphenyltetrazolium bromide or MTT) to purple formazan in the cells. BV2 cells (approx. 25 × 10^3^ cells/well) were seeded on 96-well cell culture plates and incubated for 24 h in 5% CO_2_ at 37 °C. On the following day, the medium was changed and after at least 1 h, cells were pre-treated with different concentrations of X15856 for 30 min (10 and 25 μM). The cells were then incubated with or without LPS (10 ng/mL) in the absence or presence of the compound for the next 20 h. In total, 20 μL of 100% Ethanol (approximately 20% end conc.) was used as a positive control to induce cell death. Next, 20 μL of MTT solution (working concentration of 5 mg/mL) was added to all wells, followed by another 4 h incubation at 37 °C. Afterwards, the medium was removed and replaced by 200 μL of DMSO to detect the intracellular formazan crystals formed in the viable cells per well. Lastly, the colorimetric reaction was measured using an MRX^e^ Microplate reader (Dynex Technologies, Denkerdorf, Germany) at photometric extinction at a 570 nm wavelength with a reference wavelength of 630 nm [88,89].

### 4.4. Determination of Cytokine and Chemokine Release

ELISA is used for quantitative protein analysis directly in adherent cell’s supernatant. BV2 microglial cells were left untreated or pre-treated with different concentrations of X15856. After that, LPS was added for 24 h except for the negative control. Afterwards, the supernatants were collected and centrifuged at 1000× *g* for 2 min at 4 °C. Commercially available ELISA kits (R&D Systems Europe, Ltd., Abingdon, UK) were used to investigate the effects of X15856 (1 μM to 25 μM), dexamethasone, and hydrocortisone (0.1, 1, and 10 μM) on the protein synthesis of TNF-α, IL-6, CCL2, CCL3, CXCL2, and CXCL10 as instructed by the manufacturer. Briefly, 96-well plates (Thermo Fisher Scientific, Bonn, Germany) were coated with the respective capture antibodies. On the following day, the plates were blocked and washed, and standards and supernatants were prepared based on the pre-test-determined dilutions and added into the respective wells. Detection antibody, streptavidin-HRP, and stop solution were added step by step as advised by the manufacturer. The absorbance of the wells was read at 450 nm using the MRXe Microplate reader. Data were normalized to LPS control and presented as a percentage of change in cytokine and chemokine levels.

### 4.5. RNA Isolation and Quantitative PCR

For the quantification of the gene expression of the various inflammatory parameters, we performed quantitative real-time PCR (qPCR) in BV2 microglial cells. Cultured cells were pre-treated with the X15856 (1 to 25 μM) for 30 min, followed by stimulation with LPS (10 ng/mL) for 4 h. RNA preparation was performed using a GeneMATRIX Universal RNA Purification Kit (Roboklon GmbH, Berlin, Germany), according to the manufacturer’s protocol. Then, cDNA was reverse transcribed from 500 ng of total RNA using MMLV reverse transcriptase and random hexamers (biomers.net GmbH, Ulm, Germany). The synthesized cDNA was used as the template for the real-time qPCR amplification carried out by the CFX96 real-time PCR detection system (Bio-Rad Laboratories GmbH, Feldkirchen, Germany) using SYBR Green supermix (Bio-Rad Laboratories GmbH, Munich, Germany). Glyceraldehyde 3-phosphate dehydrogenase (GAPDH) served as an internal control for sample normalization. The primer sequences were GAPDH: Forward (Fwd): 5′-TGGGAAGCTGGTCATCAAC-3′/Reverse (Rev): 5′-GCATCACCCCATTTGATGTT-3′, TNF-α: Fwd: 5′-CCCACGTCGTAGCAAACCACCA-3′/Rev: 5′-CCATTGGCCAGGAGGGCGTTG-3′, IL-6: Fwd: 5′-AGTTGCCTTCTTGGGACTGA-3′/Rev: 5′-TTCTGCAAGTGCATCATCGT-3′, CCL2: Fwd: 5′-TGATCCCAATGAGTAGGCTGG-3′/Rev: 5′-ACCTCTCTCTTGAGCTTGGTG-3′, CCL3: Fwd: 5′-TATTTTGAAACCAGCAGCCTTT-3′/Rev: 5′-ATTCTTGGACCCAGGTCTCTTT-3′, CXCL2: Fwd: 5′-CCCTCAACGGAAGAACCAAAG 3′/Rev: 5′-GAGGCACATCAGGTACGATCCA 3′, and CXCL10: Fwd: 5′-CAGTGGATGGCTAGTCCTAATTG-3′/Rev: 5′-ACTCAGACCAGCCCTTAAAGAAT-3′. Specific primers were designed using Primer-BLAST and obtained with biomers.net GmbH (Ulm, Germany).

### 4.6. Immunoblotting

BV2 cells were stimulated with LPS (10 ng/mL) with or without X15856 (1 to 25 μM) for 30 min. Afterwards, cells were washed with cold PBS and lysed in lysis buffer (42 mM Tris-HCl, 1.3% sodium dodecyl sulfate, 6.5% glycerin, 100 μM sodium orthovanadate, and 2% phosphatase and 0.2% protease inhibitors). The bicinchoninic acid (BCA) protein assay kit (Thermo Fisher Scientific, Bonn, Germany) was used for protein concentration estimation according to the manufacturer’s instructions and evaluated using a microplate reader at 570 nm. Before electrophoresis, bromophenol blue and DTT (10 mM each) were added to the samples and 20 μg of the total protein from each sample was subjected to sodium dodecyl sulfate-polyacrylamide gel electrophoresis (SDS-PAGE) under reducing conditions. The proteins were then transferred to 0.45 μm polyvinylidene fluoride (PVDF) membranes (Merck Millipore, Darmstadt, Germany) by semi-dry blotting. The membranes were blocked overnight using Roti-Block (Roth, Karlsruhe, Germany), and the following day, the membranes were incubated for at least 2 h with primary antibodies at room temperature. Primary antibodies were rabbit anti-phospho-PKC (pan) (βII Ser660) (1:1000; Cell Signaling Technology, Frankfurt, Germany), rabbit anti-ERK1/2 (1:1000; Cell Signaling Technology, Frankfurt, Germany), rabbit anti-p38 MAPK (1:1000; Cell Signaling Technology, Frankfurt, Germany), rabbit anti-phospho-NF-kappaB-p65 (1:1000; Cell Signaling Technology, Frankfurt, Germany), and mouse anti-Vinculin (1:20,000, Merck, Darmstadt, Germany). After the washing of the membranes, protein-bound antibodies were detected with horseradish peroxidase-coupled goat anti-rabbit IgG (1:10,000, SeraCare Life Sciences Inc., Milford, MA, USA) and mouse anti-rabbit IgG (1:10,000 EMD Millipore, Darmstadt, Germany), using enhanced chemiluminescence (ECL) reagents (Biozym, Hessisch Oldendorf, Germany). Pictures were taken using the ChemiDoc MP imaging system (Bio-Rad Laboratories GmbH, Feldkirchen, Germany). The equality of the protein loading and transfer was evaluated by subjecting each sample to a Western blot for Vinculin protein for signaling. The densitometric analysis was performed using ImageJ software (V1.48t, NIH, Stapleton, NY, USA).

### 4.7. Statistical Analysis

All statistical analyses were carried out using GraphPad Prism 8.0 (Prism 8 software, GraphPad software Inc., San Diego, CA, USA). Raw values were converted to the percentage of LPS (10 ng/mL) or the appropriate positive control, such as untreated cells for MTT assay, and the appropriate control was considered 100%. The data are represented as the mean ± SEM of at least three independent experiments. Multiple comparisons data were analyzed using one-way ANOVA with Dunnett’s post hoc test. The level of significance was set at * *p* < 0.05, ** *p* < 0.01, *** *p* < 0.001, and **** *p* < 0.0001 and is indicated in the figures.

## 5. Conclusions

Neuroinflammation is a key factor that might be part of the pathogenesis of psychiatric and neurological disorders. However, effective pharmacological therapies remain to be fully characterized. In this study, we demonstrated that the hexacyclic peptide-peptoid hybrid X15856 exerts anti-neuroinflammatory effects in BV2 microglial cells by suppressing the LPS-induced expression and release of cytokines and chemokines. The observed effects were partially dependent on the modulation of MAPK, PKC, and NF-κB signaling pathways as determined by their phosphorylation levels. Therefore, peptoids and peptoid-peptide hybrids display high potential as anti-neuroinflammation drugs in the treatment of CNS diseases such as depression and should be further investigated.

## Figures and Tables

**Figure 1 ijms-25-04462-f001:**
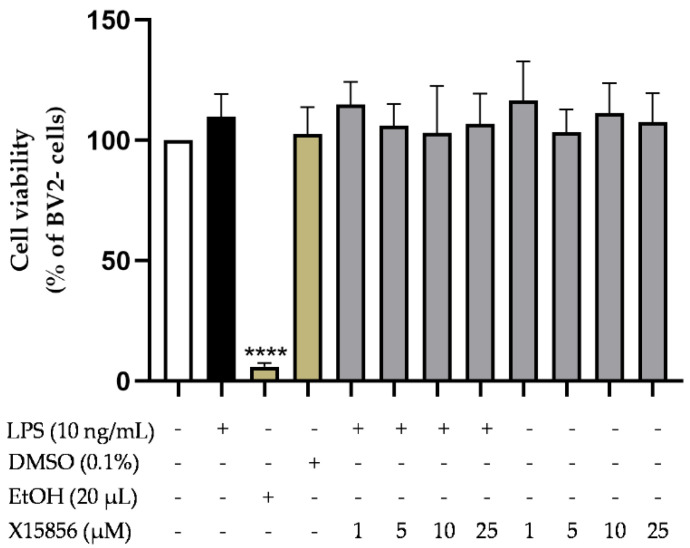
Effects of X15856 on cell viability of LPS-stimulated BV2 microglial cells. Cell viability was measured by color change due to MTT reduction after 24 h of treatment. Values are presented as the mean ± SD of three independent experiments. Statistical analysis was performed using one-way ANOVA with Dunnett’s post hoc test with **** *p* < 0.0001 compared to untreated cells.

**Figure 2 ijms-25-04462-f002:**
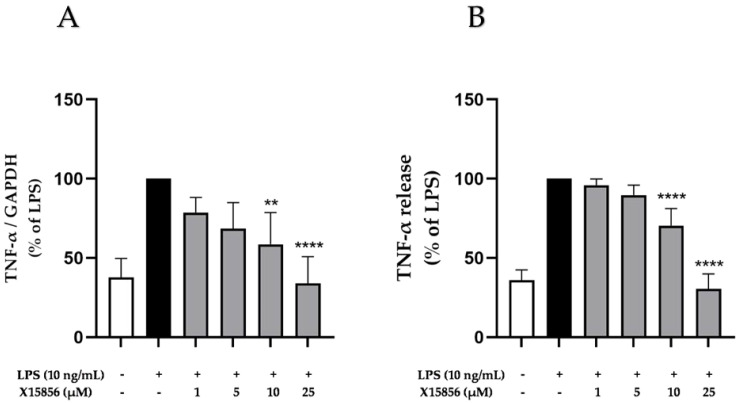
Effects of X15856 on the gene expression (**A**) and protein synthesis (**B**) of TNF-α in LPS-stimulated BV2 cells. Cells were stimulated as described in the Materials and Methods Section. (**A**) After 4 h of stimulation, RNA was isolated and the gene expression of TNF-α was measured using qPCR. (**B**) After 24 h of stimulation, supernatants were collected and the release of TNF-α was measured with ELISA. Values are presented as the mean ± SD of at least three independent experiments. Statistical analysis was performed using one-way ANOVA with Dunnett’s post hoc tests with ** *p* < 0.01 and **** *p* < 0.0001 compared to LPS.

**Figure 3 ijms-25-04462-f003:**
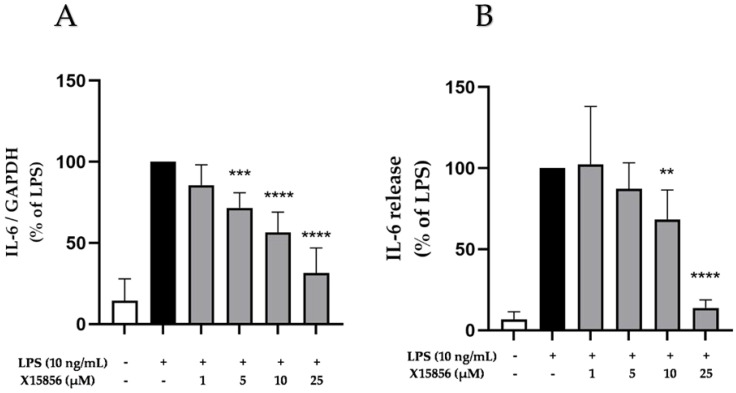
Effects of X15856 on the gene expression (**A**) and protein synthesis (**B**) of IL-6 in LPS-stimulated BV2 cells. Cells were stimulated as described in the Materials and Methods Section. (**A**) After 4 h of stimulation, RNA was isolated and gene expression of IL-6 was measured using qPCR. (**B**) After 24 h of stimulation, supernatants were collected, and the release of IL-6 was measured with ELISA. Values are presented as the mean ± SD of at least three independent experiments. Statistical analysis was performed using one-way ANOVA with Dunnett’s post hoc tests with ** *p* < 0.01 *** *p* < 0.001 and **** *p* < 0.0001 compared to LPS.

**Figure 4 ijms-25-04462-f004:**
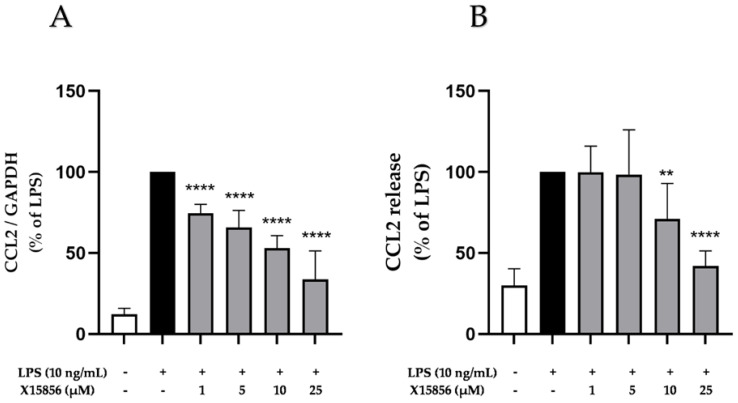
Effects of X15856 on the gene expression (**A**) and protein synthesis (**B**) of CCL2 in LPS-stimulated BV2 cells. Cells were stimulated as described in the Materials and Methods Section. (**A**) After 4 h of stimulation, RNA was isolated and the gene expression of CCL2 was measured using qPCR. (**B**) After 24 h of stimulation, supernatants were collected and the release of CCL2 was measured with ELISA. Values are presented as the mean ± SD of at least three independent experiments. Statistical analysis was performed using one-way ANOVA with Dunnett’s post hoc tests with ** *p* < 0.01 and **** *p* < 0.0001 compared to LPS.

**Figure 5 ijms-25-04462-f005:**
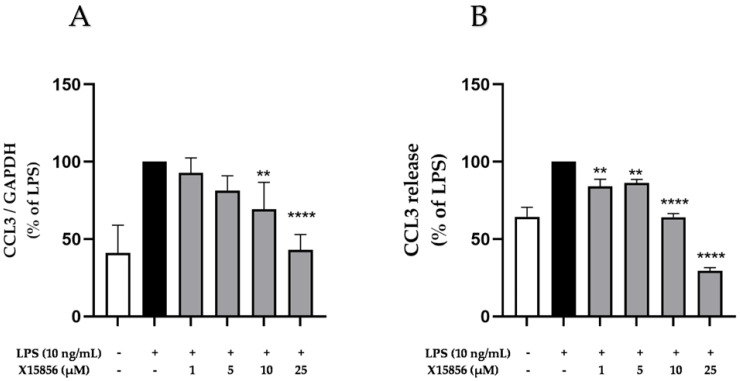
Effects of X15856 on the gene expression (**A**) and protein synthesis (**B**) of CCL3 in LPS-stimulated BV2 cells. Cells were stimulated as described in the Materials and Methods Section. (**A**) After 4 h of stimulation, RNA was isolated and the gene expression of CCL3 was measured using qPCR. (**B**) After 24 h of stimulation, supernatants were collected and the release of CCL3 was measured with ELISA. Values are presented as the mean ± SD of at least three independent experiments. Statistical analysis was performed using one-way ANOVA with Dunnett’s post hoc tests with ** *p* < 0.01, and **** *p* < 0.0001 compared to LPS.

**Figure 6 ijms-25-04462-f006:**
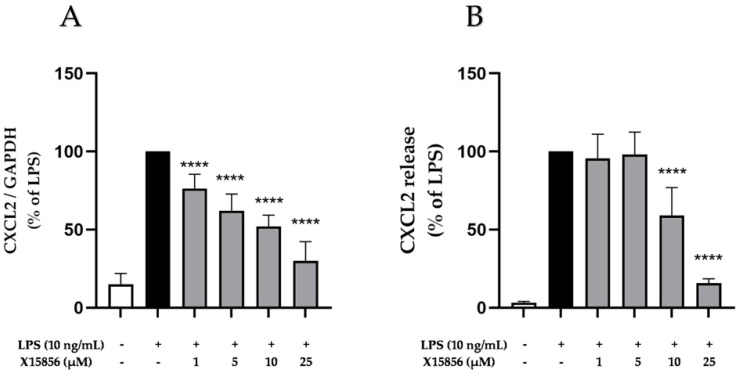
Effects of X15856 on the gene expression (**A**) and protein synthesis (**B**) of CXCL2 in LPS-stimulated BV2 cells. Cells were stimulated as described in the Section 4. (**A**) After 4 h of stimulation, RNA was isolated and the gene expression of CXCL2 was measured using qPCR. (**B**) After 24 h of stimulation, supernatants were collected and the release of CXCL2 was measured with ELISA. Values are presented as the mean ± SD of at least three independent experiments. Statistical analysis was performed using one-way ANOVA with Dunnett’s post hoc tests with **** *p* < 0.0001 compared to LPS.

**Figure 7 ijms-25-04462-f007:**
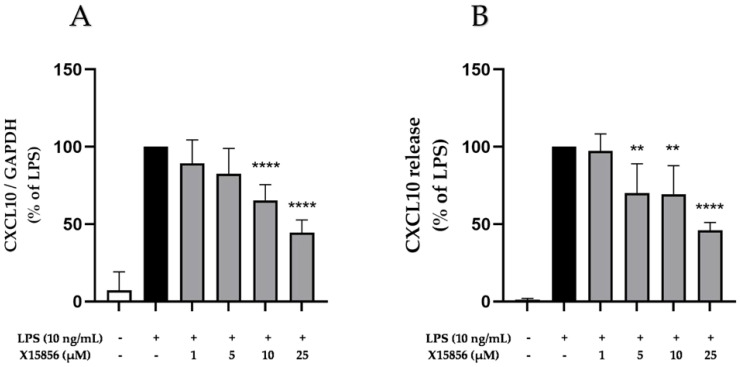
Effects of X15856 on the gene expression (**A**) and protein synthesis (**B**) of CXCL10 in LPS-stimulated BV2 cells. Cells were stimulated as described in the Materials and Methods Section. (**A**) After 4 h of stimulation, RNA was isolated and the gene expression of CXCL10 was measured using qPCR. (**B**) After 24 h of stimulation, supernatants were collected and the release of CXCL10 was measured with ELISA. Values are presented as the mean ± SD of at least three independent experiments. Statistical analysis was performed using one-way ANOVA with Dunnett’s post hoc tests with ** *p* < 0.01 and **** *p* < 0.0001 compared to LPS.

**Figure 8 ijms-25-04462-f008:**
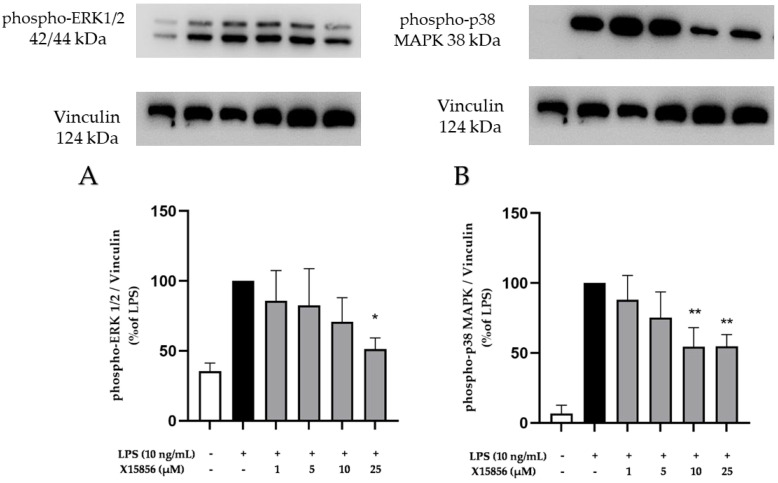
Effects of X15856 on the phosphorylation of ERK 1/2 (**A**) and p38 MAPK (**B**) in LPS-stimulated BV2 cells. Cells were stimulated as described in the Materials and Methods Section. Values are presented as the mean ± SD of at least three independent experiments, and protein levels were referenced to Vinculin. Statistical analysis was performed using one-way ANOVA with Dunnett’s post hoc tests with * *p* < 0.05 and ** *p* < 0.01 compared to LPS.

**Figure 9 ijms-25-04462-f009:**
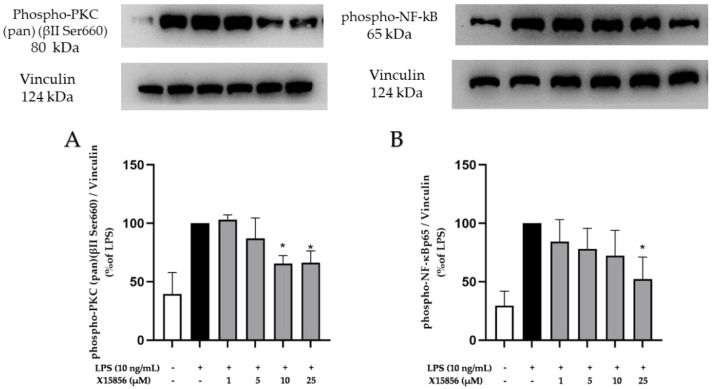
Effects of X15856 on the phosphorylation of PKC (pan) (βII Ser660) (**A**) and NF-κB (**B**) in LPS-stimulated BV2 cells. Cells were stimulated as described in the Material and Methods Section. Values are presented as the mean ± SD of at least three independent experiments, and protein levels were referenced to Vinculin. Statistical analysis was performed using one-way ANOVA with Dunnett’s post hoc tests with * *p* < 0.05 compared to LPS.

**Figure 10 ijms-25-04462-f010:**
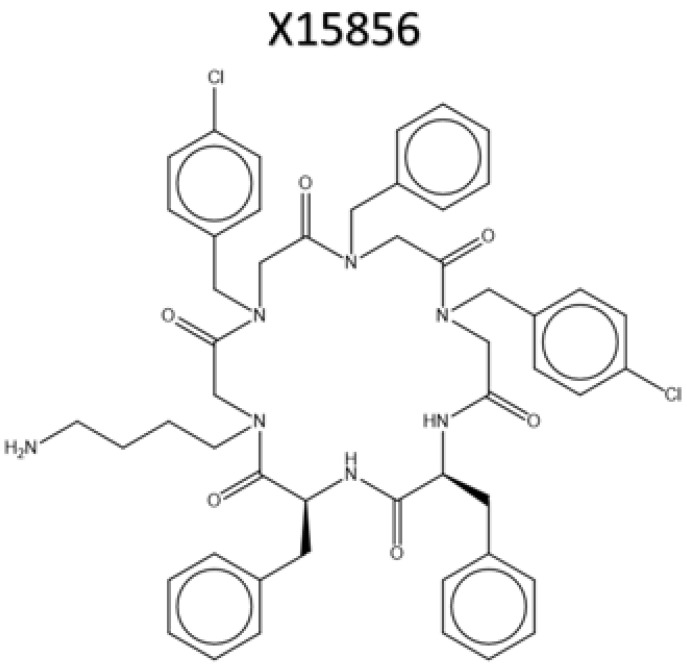
Molecular structure of the synthesized compound X15856. A hexacyclic hybrid of two peptide units and four peptoid units.

## Data Availability

The data presented in this manuscript are available from the corresponding author upon request. All data on the synthesis and analysis of the investigated compound X15856 are available via the Chemotion repository (https://www.chemotion-repository.net/, (accessed on 15 February 2024)) and can be accessed via the following collection DOI: https://doi.org/10.14272/collection/KA_2023-11-13 [90].

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
