# Peer review of "Anti-Neuroinflammatory Effects of a Macrocyclic Peptide-Peptoid Hybrid in Lipopolysaccharide-Stimulated BV2 Microglial Cells"

_ijms, 2024, doi:10.3390/ijms25084462_

Round 1
Reviewer 1 Report
Comments and Suggestions for Authors
This manuscript investigated the anti-neuroinflammatory potential of a novel compound (X15856) in LPS-stimulated microglial cells. The molecule mitigated the production of cytokines and chemokines evoked by LPS exposure in BV2 cells. Additionally, the authors showed that these effects may be attributed to the modulation of the MAPK, PKC, and NF-κB signaling pathways.
Altogether, the results point out that the compound presents an anti-inflammatory effect in vitro and could be further used for the development of adjuvant strategies to treat illnesses with an inflammation background, such as depression. The paper is generally well-written, and the findings are interesting and add to the field. I recommend some amendments to improve the work before publication.
Q1. The title and the abstract should disclose that the work was performed in vitro. The information that cells were challenged with LPS could also be presented in the title.
Q2. I would recommend the use of a known anti-inflammatory drug for comparison, but this is not mandatory.
Q3. There is a need in this study for an additional protocol to complement the MTT assay. The results of this technique should be accompanied by another assay (CCK-8, for example) to evaluate cytotoxicity/proliferation.
Also, the wavelength reported for this protocol is usually 570 nm, and in the text is 595 nm, please check this issue. Although the X15856 compound did not present cytotoxic effects in higher concentrations, the MTT initial curve should include all the concentrations used for later experiments, i.e., 1, 5, 10, and 25 µM.
Q4. The reference for the concentration of LPS (10 ng/mL) used in this paper should be cited in the text.
Q5. I suggest the authors group the results and figures from the qPCR and ELISA protocols. Many figures in the article could be placed together, such as the results from the cytokines (IL-6 and TNF-α) and chemokines (CCL2, CCL3, CXCL2, and CXCL10). This would make the description of the results more concise. In Figure 5, I believe the authors mean CCL3 in the description of the Y-axis, please check.
Q6. Why did the authors choose the protein vinculin as the internal control for western blot analysis? The most common ones are tubulin, gapdh, or actin. Also, check the spelling of the protein vinculin which is incorrect in western blot figures (8 and 9).
Q7. Some of the caveats of the study should be discussed by the authors.
Author Response
"Please see the attachment."

Reviewer 2 Report
Comments and Suggestions for Authors
Dear Authors,
You present here the investigation of the anti-neuroinflammatory effects a new peptide-peptoid hybrid in BV2 microglial cells.
After carefully reading your manuscript, I have the above suggestions:
1. you should insert the structure of the compound X15856 in the Results section, not in Material and methods
2. the figures are not easy to understand; for example, what does the columns represent? the concentrations are not mentioned. please redo the figures
3. reduce the letter size for A and B in figures
4. there is no reference drug used in the tests
5. I suggest you do a structure-activity relationship for the compound
6. there are some minor English corrections, such as the add of "the" before many terms; ex. "after the treatment" (line 24), "we report the.." (line 26), "by the modulation" (line 31); "by the prevention" (line 61); "the effects"(line 104), etc.
7. the references are well chosen and in the agreement with the subject
Comments on the Quality of English Languagethere are some minor English corrections, such as the add of "the" before many terms; ex. "after the treatment" (line 24), "we report the.." (line 26), "by the modulation" (line 31); "by the prevention" (line 61); "the effects"(line 104), etc.
Author Response
"Please see the attachment."

Reviewer 3 Report
Comments and Suggestions for Authors
This manuscript provides new insights on the role of macroglia and the use of molecules able to modulate microglia activity.
I only have few comments that need to be addressed:
Introduction:
I would suggest to provide more background about microglia, and why these cells are important.
Fig 1: I would suggest to add 2 other conditions: 1- cells + molecule 10uM
2- cells + the molecule 25uM since you want to evaluate the toxicity of the molecule, this conditions should be added
Same comment for other figures. It is important to see the baseline effect of you compound on cells.
On the graph the Y axis you mention (% of LPS) what does it mean? does it mean the LPS condition is you reference ?
Out of curiosity, why using Vinclin to normalize instead of Actin ?
Discussion:
Please could you discuss the limits of your study?
It would be interesting to discuss the apparent dose effect in this section.
Last but not least, I would suggest to harmonize colors and shape on the graphs.
Thanks
Author Response
"Please see the attachment."

Round 2
Reviewer 2 Report
Comments and Suggestions for Authors
Dear authors,
Thank you for considering some of my comments and for making the changes that I suggested.
Still, I consider that every graphic has to have the x and y axes well defined, with the parameter observed and the measure units.
For the anti-inflammatory assay, I still suggest that you need to add a reference drug.
Author Response
"Please see the attachment."

Reviewer 3 Report
Comments and Suggestions for Authors
Thank you for the modifications
Author Response
"Please see the attachment."
